# Learning Topics using Semantic Locality

**Ziyi Zhao , Krittaphat Pugdeethosapol , Sheng Lin , Zhe Li , Yanzhi Wang , Qinru Qiu**
Department of Electrical Engineering & Computer Science
Syracuse University
Syracuse, NY 13244, USA
{zzhao37, kpugdeet, shlin, zli89, ywang393, qiqiu}@syr.edu

## Abstract

The topic modeling discovers the latent topic probability of the given text documents. To generate the more meaningful topic that better represents the given document, we proposed a new feature selection technique which can be used in the data preprocessing stage. The method consists of three steps. First, it generates the word/word-pair from every single document (Feature generation). Second, it applies a two-way TF-IDF algorithm to word/word-pair for semantic filtering (Feature filtering). Third, it uses the K-means algorithm to merge the word pairs that have the similar semantic meaning (Feature coalescence). Our proposed technique can improve the generated topic accuracy by up to 12.99%.

## 1 Introduction

Topic modeling is a collection of algorithms that aim to discover and annotate large archives of documents with thematic information(Blei, 2012). It enables us to convert a collection of large documents into a set of topic vectors. Each entry in this concise representation is a probability of the latent topic distribution. By comparing the topic distributions, we can easily calculate the similarity between two different documents(Steyvers & Griffiths, 2007). The availability of many manually categorized online documents, such as Internet Movie Database (IMDb) movie review (Inc., 1990), Wikipedia articles, makes the testing and validation of topic models possible.

Some topic modeling algorithms are highly frequently used in text-mining(Mei et al., 2008), preference recommendation(Wang & Blei, 2011) and computer vision(Wang & Grimson, 2008). Many of the traditional topic models focus on latent semantic analysis with unsupervised learning (Blei, 2012). The Probabilistic Latent Semantic Analysis (PLSA)(Hofmann, 1999) model uses maximum likelihood estimation to extract latent topics and topic word distribution, while the Latent Dirichlet Allocation (LDA) (Blei et al., 2003) model performs iterative sampling and characterization to search for the same information. Restricted Boltzmann Machine (RBM) (Hinton & Salakhutdinov, 2009) is also a very popular model for the topic modeling. By training a two layer model, the RBM can learn to extract the latent topics in an unsupervised way.

In this paper, we will explore the existing knowledge and build a topic model using explicit semantic analysis. This work studies effective data processing and feature extraction for topic modeling and information retrieval. We investigate how the available semantic knowledge, which can be obtained from language analysis, can assist in the topic modeling.

## 2 Approach

Our feature selection contains three steps handled by three different modules: feature generation module, feature filtering module, and feature coalescence module. The overall structure of our framework is shown in Figure 1(a). The proposed feature selection is based on our observation that word dependencies provide additional semantic information than simple word counts. However, there are quadratically more depended word pairs relationships than words. To avoid the explosion of feature set, filtering and coalescing must be performed.

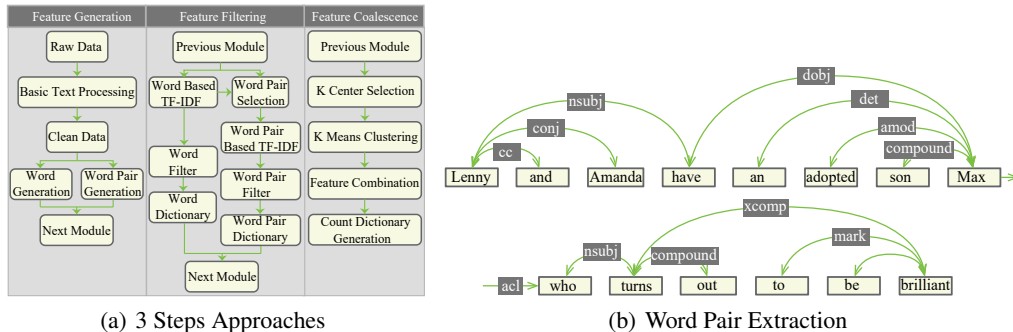

(a) 3 Steps Approaches            (b) Word Pair Extraction

Figure 1: Model structure

## 2.1 FEATURE GENERATION: SEMANTIC WORD PAIR EXTRACTION

Current RBM model for topic modeling uses the bag-of-words approach. Each visible neuron represents the number of appearance of a dictionary word. Our hypothesis is that including word pairs (with specific dependencies) helps to improve topic modeling.

In this work, Stanford natural language parser (Chen & Manning, 2014) is used to analyze sentences in both training and testing corpus, and extract word pairs that are semantically dependent. For example, given the sentence: *"Lenny and Amanda have an adopted son Max who turns out to be brilliant."*. Figure 1(b) shows all the depended word pairs extracted using the Standford parser. As you can see that the depended words are not necessarily adjacent to each other, however they are semantically related.

## 2.2 FEATURE FILTERING: TWO STEPS TF-IDF PROCESSING

The word and word pair dictionaries still contain a lot of high frequency words that are not very informational, such as "first", "name", etc. *Term frequency-inverse document frequency (TF-IDF)* is applied to screen out those unimportant words or word pairs and keep only important ones.

As shown in Figure 1(a), Feature Filtering module, a two-step TF-IDF processing is adopted. First, the word-level TF-IDF is performed. The result of word level TF-IDF is used as a filter and a word pair is kept only if the TF-IDF scores of both words are higher than the threshold (0.01). After that, we treat each word pair as a single unit, and the TF-IDF algorithm is applied again to the word pairs and further filter out word pairs that are either too common or too rare. Finally, this module will generate the filtered word and word pair dictionaries.

## 2.3 FEATURE COALESCENCE: K-MEANS CLUSTERING

Even with the TF-IDF processing, the size of the word pair dictionary is still prohibitively large. We further cluster semantically close word pairs to reduce the dictionary size. Each word is represented by their embedded vectors calculated using Google's word2vec model. The semantic distance between two words is measured as the Euclidean distance of their embedding vectors. The words that are semantically close to each other are grouped into K clusters.

We use the index of each cluster to replace the words in the word pair. If the cluster ID of two word pairs are the same, then the two word pairs are semantically similar and be merged. In this we can reduce the number of word pairs by more than 63%.

## 3 EVALUATION

The proposed topic model will be tested in the context of content-based recommendation. Given a query document, the goal is to search the database and find other documents that fall into the category by analyzing their contents. In our experiment, we generate the topic distribution of each document by using RBM model. Then we retrieve the top N documents whose topic is the closet to the query document by calculating their Euclidean distance. Our proposed method is evaluated on 3 datasets: OMDb, Reuters, and 20NewsGroup. And we use $mean\ Average\ Precision\ (mAP)$ score to evaluate our proposed method.

Table 1: Fixed total feature number word/word pair performance evaluation

| mAP | F = 10.5K | | F = 11K | | F = 11.5K | | F = 12K | | F = 12.5K | | F = 15K | |
|---|---|---|---|---|---|---|---|---|---|---|---|---|
| | word | word pair | word | word pair | word | word pair | word | word pair | word | word pair | word | word pair |
| **OMDB** | | | | | | | | | | | | |
| mAP 1 | **0.14772** | 0.14603 | 0.13281 | **0.14673** | 0.13817 | **0.14789** | 0.13860 | **0.14754** | 0.14019 | **0.14870** | 0.13686 | **0.14708** |
| mAP 3 | 0.09381 | **0.09465** | 0.08606 | **0.09327** | 0.08933 | **0.09507** | 0.08703 | **0.09517** | 0.09054 | **0.09657** | 0.09009 | **0.09537** |
| mAP 5 | 0.07453 | **0.07457** | 0.06835 | **0.07380** | 0.07089 | **0.07508** | 0.06925 | **0.07485** | 0.07117 | **0.07635** | 0.07175 | **0.07511** |
| mAP 10 | 0.05273 | **0.05387** | 0.04862 | **0.05340** | 0.04976 | **0.05389** | 0.04900 | **0.05322** | 0.05019 | **0.05501** | 0.05083 | **0.05388** |
| **Reuters** | | | | | | | | | | | | |
| mAP 1 | 0.94195 | **0.95127** | 0.94277 | **0.95023** | 0.94407 | **0.95179** | 0.94244 | **0.94997** | 0.94277 | **0.95270** | 0.94163 | **0.94984** |
| mAP 3 | 0.92399 | **0.93113** | 0.92448 | **0.93117** | 0.92604 | **0.93276** | 0.92403 | **0.93144** | 0.92249 | **0.93251** | 0.92326 | **0.93353** |
| mAP 5 | 0.91367 | **0.92123** | 0.91366 | **0.91939** | 0.91589 | **0.92221** | 0.91367 | **0.92051** | 0.91310 | **0.92063** | 0.91284 | **0.92219** |
| mAP 10 | 0.89813 | **0.90425** | 0.89849 | **0.90296** | 0.90050 | **0.90534** | 0.89832 | **0.90556** | 0.89770 | **0.90365** | 0.89698 | **0.90499** |
| **20NewsGroup** | | | | | | | | | | | | |
| mAP 1 | 0.73736 | **0.77129** | 0.73375 | **0.76093** | 0.68720 | **0.75865** | 0.73959 | **0.75846** | 0.72280 | **0.76768** | 0.72695 | **0.75583** |
| mAP 3 | 0.65227 | **0.68905** | 0.64848 | **0.68042** | 0.60356 | **0.67546** | 0.65530 | **0.67320** | 0.63649 | **0.68455** | 0.63951 | **0.66743** |
| mAP 5 | 0.60861 | **0.64620** | 0.60548 | **0.63783** | 0.56304 | **0.63321** | 0.61115 | **0.62964** | 0.59267 | **0.64165** | 0.59447 | **0.62593** |
| mAP 10 | 0.55103 | **0.58992** | 0.54812 | **0.58057** | 0.51188 | **0.57839** | 0.55338 | **0.57157** | 0.53486 | **0.58500** | 0.53749 | **0.56969** |

### 3.1 WORD/WORD PAIR PERFORMANCE COMPARISON

In this experiment, we compare the performance of two RBM models. One of them only considers words as the input feature, while the other has combined words and word pairs as the input feature. The total feature size varies from 10500, 11000, 11500, 12000, 12500, 15000. For the word/word pair combined RBM model, the number of word feature is fixed to be 10000, and the number of word pair features is set to meet the requirement of total feature size.

Both models are applied to all three datasets, and the results are shown in Table 1, the word/word pair combined model almost always performs better than the word-only model. For the $mAP1$, the $mAP3$, the $mAP5$ and the $mAP10$ up to 10.48%, 11.91%, 12.46%, and 12.99% improvement are achieved.

### 3.2 WORD PAIR GENERATION PERFORMANCE

In this experiment, we compare different word pair generation algorithms with the baseline. Similar to previous experiments, the baseline is the word-only RBM model whose input consists of the 10000 most frequent words. The "Semantic" word pair generation is the method we proposed in this paper. We use the idea from the skip-gram (Mikolov et al., 2013) that has a window size of N = 2 as "N-gram". For the "Non-K", we use the same algorithm as the "Semantic" except that no K-means clustering is applied on the generated word pairs.

Table 2: Different word pair generation algorithms for OMDb

| mAP | Baseline | Semantic | N-gram | Non-K |
|---|---|---|---|---|
| mAP 1 | 0.14134 | **0.14870** | 0.13202 | 0.14302 |
| mAP 3 | 0.09212 | **0.09657** | 0.08801 | 0.09406 |
| mAP 5 | 0.07312 | **0.07635** | 0.07111 | 0.07575 |
| mAP 10 | 0.05113 | **0.05501** | 0.05132 | 0.05585 |

There are several things we can observe from the Table 2. First, the "semantic" word pair generation gives us the best $mAP$ score. This is because, although both "Non-K" and "semantic" techniques extract word pairs using natural language processing, without the K-means clustering, semantically similar pairs will be considered separately. Hence there will be lots of redundancies in the input space. The K-means clustering performs the function of compression and feature extraction. Second, the "N-gram" word pair generation, its $mAP$ score is even lower than the baseline. This is because the "N-gram" word pair generation simply extracts words that are adjacent to each other, without the semantic meanings and grammatical dependencies. When introducing some meaningful word pairs, it also introduces more meaningless word pairs at the same time. These meaningless word pairs act as noises in the input and does not help to improve the model accuracy.

## 4 CONCLUSION

In this paper, we proposed techniques to preprocess the dataset and optimize the original RBM model. We replaced the original word only RBM model by introducing word pairs. Then, we showed that proper selection of word pair generation techniques can significantly improve the topic prediction accuracy and the document retrieval performance. With our improvement, experimental results have verified that, compared to original word only RBM model, our proposed word/word pair combined model can improve the $mAP$ score up to 10.48% in OMDb dataset, up to 1.11% in Reuters dataset and up to 12.99% in the 20NewsGroup dataset.

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
