# OpenReview forum: "Learning Topics Using Semantic Locality"
_ICLR.cc/2018/Workshop — Reject_

### Official Review · AnonReviewer2 · 2018-03-05
**many details missing, limited technical contribution**

**Rating:** 3
**Confidence:** 4

**Review:**

This paper leverages Stanford parser to extract dependencies between words to generate word-pairs that are not simply co-occurring word pairs and propose this as a feature selection technique before applying a K-means algorithm to group/merge words pairs with similar meaning. Experiments with/without these features using an RBM model are shown to demonstrate improved mean average precision scores. Missing important details, and limited technical contribution are some of the major shortcomings of this paper.

- The example in section 2.2 seems to be context-less. "first", "name" both of these words can be important in a sentence like "My first name is ...".

- How did you determine the threshold of 0.01 for TF-IDF scores?

- What is the value of K? How did you determine this value?

- No details of the datasets are provided.

- How did you vary the feature size? Why did you fix the feature size to 10000?

- Section 3.2 is interesting, but some examples and more analyses could be helpful. How does your semantic model compare with the N-gram model if N > 2?

- Why do you think there is a minor improvement of 1.11% for the Reuters dataset?

- Since this paper claims to add value in the topic modeling domain, comparisons need to be shown with the state-of-the-art topic models on the same datasets to demonstrate how the proposed feature selection technique is useful.

---

### Official Review · AnonReviewer3 · 2018-03-10
**More rigorous experimentation is required**

**Rating:** 3
**Confidence:** 5

**Review:**

This workshop submission was revised from a previous-rejected full ICLR18 submission.

Many major issues exist in the submission and have been pointed out clearly by
https://openreview.net/forum?id=Byni8NLHf&noteId=HybCHS8Hz

For instance:
1. Does using linked-word-pairs truly improve the state of the art? The experimental results only compare RBMs with and without the added pair features.

2. Are word pairs from dependency-parsing truly better than co-occurring words? If yes, Why? If yes, is this improvement only about topic modeling?

3. The idea of using word-pairs is not new. The bigger issue is the high-dimensional challenge. The submission used multiple standard strategies but providing no insights of why.

---

### Official Review · AnonReviewer1 · 2018-03-20
**not very interesting**

**Rating:** 4
**Confidence:** 4

**Review:**

This paper proposes a pipeline for topic modeling. The proposed pipeline consists three different steps for feature generation, filtering, and coalescence. I think the proposed pipeline is overcomplex and lack of motivation. The followings are my questions:
1. It is unclear to me in this paper why the authors want to use the parse tree to define the word pairs. How does parse tree help topic modeling?
2. I think the Stanford neural-based parser uses word embedding during training. Why the authors need another set of word embedding in section 2.3?
3. The word2vec word embeddings are good for syntactically equivalent measurements. It does not work well for semantic comparison. For example, the nearest neighbor for "good" is "bad" in word2vec. More examples can be found in Kim's CNN paper ("Convolutional Neural Networks for Sentence Classification"). Therefore, the semantic distance measurement is not reliable to me in Sec. 2.3.

The pipeline which is proposed in this paper is very complex to me and it is also not an end-to-end system. It is hard to tell which part of the framework plays a more important role in the model. There are also a lot of parameters for each component. I don't recommend to accept this paper.

---

### Decision · Program_Chairs · 2018-03-20
**ICLR 2018 Workshop Acceptance Decision**

**Decision:**

Reject

**Comment:**

Based on the reviews, this paper has not been accepted for presentation at the ICLR workshop. However, the conversation and updates can continue to appear here on OpenReview.